# ‘The Little Engine That Could’: A Qualitative Study of Medical Service Access and Effectiveness among Adolescent Athletics Athletes Competing at the Highest International Level

**DOI:** 10.3390/ijerph18147278

**Published:** 2021-07-07

**Authors:** Toomas Timpka, Kristina Fagher, Victor Bargoria, Håkan Gauffin, Christer Andersson, Jenny Jacobsson, James Nyce, Stéphane Bermon

**Affiliations:** 1Athletics Research Center, Linköping University, 58183 Linköping, Sweden; kristina.fagher@med.lu.se (K.F.); bargoriavictor@gmail.com (V.B.); hakan.gauffin@liu.se (H.G.); christer.andersson@regionostergotland.se (C.A.); jenny.jacobsson@liu.se (J.J.); jnyce@rocketmail.com (J.N.); 2Department of Health, Medicine and Caring Sciences, Linköping University, 58183 Linköping, Sweden; 3Rehabilitation Medicine Research Group, Department of Health Sciences, Lund University, 22100 Lund, Sweden; 4Department of Orthopaedics and Rehabilitation, Moi University, Eldoret 30107, Kenya; 5Department of Anthropology, Ball State University, Muncie, IN 47306, USA; 6Health and Science Department, World Athletics, MC 98007 Monte-Carlo, Monaco; stephane.bermon@worldathletics.org; 7Laboratoire Motricité Humaine, Expertise, Sport, Santé (LAMHESS), Université Côte d’Azur, CEDEX 03, 06205 Nice, France

**Keywords:** public health, adolescents, health systems, equity in health, athletics (track and field), qualitative research methods

## Abstract

Little is known about provision of medical services to adolescents prior to participating in international top-level sports. This study aimed to investigate experiences of medical service provision among high-level adolescent athletics (track and field) athletes from three continents. A thematic narrative analysis was applied to data collected from 14 athletes by semi-structured interviews. Although competing at the highest international level, these adolescent athletes had difficulties making sense of symptoms of ill health, especially on their own. With increasing exercise loads, the athletes’ medical support needs had extended beyond the capacity of parents and local communities. As there was no organized transfer of the responsibility for medical support to sports organizations, the athletes often had to manage their health problems by themselves. There were major variations among the adolescent athletes with regards to medical service access and quality. The services used ranged from sophisticated computer-assisted biomechanical analyses to traditional healers. Decreased exercise load was the common sports injury treatment. The results of this study demonstrate how the ethical standards underpinning youth sports as well as the equal provision of medical services to adolescents are challenged across the world. Further research on health service provision to adolescent top-level athletes is warranted.

## 1. Introduction

The 1.2 billion adolescents in the world share health service needs related to their physical, cognitive, and psychosocial growth and development [1,2]. In addition to these health services, subgroups of adolescents need support to manage more specific health issues. By adopting the Sustainable Development Goals in 2016 [3], the United Nations recognized universal health coverage as a global priority. The Global Accelerated Action for the Health of Adolescents (AA-HA!) [1] and the Global Strategy for Women’s, Children’s and Adolescents’ Health (2016–2030) [4], state that all adolescents should have a fair opportunity to attain their full health potential, and none should be disadvantaged from attaining that potential. Throughout the world, increasing numbers of adolescents participate in competitive sports, some of them with the ambition to become professional athletes or to qualify to the Olympic Games [5]. For instance, the past 25 years have in the U.S. seen an increase in student participation in high school athletics from 4 million participants 1971–1972 to 7.9 million in 2015–2016 [6]. Despite calls from the International Olympic Committee for more research [7], little is known about health service needs and use among adolescent competitive athletes [8,9].

Most studies of health service provision to adolescent athletes originate from the U.S., where 2 million injuries, 500,000 physician visits, and 30,000 hospitalizations have been reported to occur each year among high school athletes [10]. Detailed epidemiological data on young competitive athletes’ injuries from other countries are limited and often restricted to single sports [11,12,13,14]. However, a broad health evaluation among Italian Youth Olympics participants representing all sports revealed that 12% had a condition warranting medical attention (mainly cardiovascular (4.5%) or pulmonary (4.5%)) [15]. Regarding equity in healthcare provision to young athletes, a recent study in the U.S. reported that athletes in rural settings are disadvantaged by that health services lack funding and certified personnel [16]. At the global level, a study in youth athletics (track and field) showed that health screening services were unevenly distributed among adolescent athletes from high-, middle, and low-income countries [17].

Medical service stewardship refers to the functions carried out by organisations and governments as they seek to achieve their policy objectives regarding health care provision [18]. These functions address attainment of objectives such as coverage, equity, access, quality, and patients’ rights. The policies may also define the relative roles and responsibilities of the public, private and voluntary (civic) sectors in the service provision. Athletics is the largest sport at the Olympic Games. Relatively few studies have reported from health monitoring among adolescent top-level athletics athletes, but those available describe overuse injuries as a prevalent problem requiring regular medical supervision [19,20,21]. Little is known about provision of medical services to adolescent athletics athletes aiming to participate in the sport at the international top-level. An important research question is then how young athletics athletes in different regions of the world perceive their opportunities for medical support. This study therefore aimed to investigate experiences from medical service provision among high-level adolescent athletics athletes from continents (Africa, Europe, Asia) where few studies have been performed in the area. The rationale was to contribute to the development of a worldwide medical service stewardship that ensures emerging professional sportspersons equal and efficient medical support.

## 2. Materials and Methods

A thematic narrative analysis [22,23,24] was used for the study, drawing on the theoretical basis of phenomenology [25,26,27]. We reached out to a sample of young athletes competing at the international top-level in athletics. Inclusion criteria were being listed among the world top-50 U18 (under 18 years of age) athletes in an athletics event and English-speaking.

### 2.1. Participants

Participants were recruited at the U18 World Athletics Championships in Nairobi, Kenya, July 2017. Based on the expected experiential diversity in the data, the depth of data generated from each participant, and the pragmatic constraints of the project, an anticipated upper sample range that would generate sufficiently rich and multi-faceted data was estimated to be under 25 [28]. National teams from six countries, representing three continents (Africa, Europe, Asia), were each asked to name four athletes who could be invited to an interview. All athletes who were approached agreed to participate. Fourteen athletes (8 boys and 6 girls) aged 17 years originating from Africa (*n* = 6), Europe (*n* = 5), and Asia (*n* = 3) had been interviewed when an in-situ decision was made that the collected data were sufficient for addressing the research question. Their athletes’ main disciplines were a middle-distance running (*n* = 7), sprints (*n* = 4), or endurance (*n* = 3) event.

### 2.2. Data Collection

Semi-structured interviews based on story completion was used for the data collection. Story completion is a projective technique, requiring a person to tell a story following a scenario [29]. We developed two brief story stems on the topic of health management among athletes (Appendix A). The story stems were based on critical incidents [30,31]. Athletes were initially asked to think about three recent injury or illness episodes that resulted in a change in their workout and competition plans. They were then asked to describe the health problem with the longest duration of these. Follow-up questions were asked about details associated with the health service provision. Thereafter, they were prompted to describe the episode with the shortest duration, and questions about this were asked. At the end of each interview, the athletes were encouraged to describe their health management experiences in more general terms. 

The interview team consisted of one female sports physiotherapist (K.F.) and three male sports physicians (specialized in orthopaedics (C.A., H.G.) and social medicine (T.T.)). The team members were trained in qualitative interviewing and all researchers performed two pilot interviews before initiating data collection. Each interview (lasting 45–75 min) was carried out by at least two researchers, with one researcher interviewing the athlete and the other taking notes and asking follow-up questions. All interviews were conducted outside at Kenyatta Stadium during the U18 World Athletics Championships in Nairobi, Kenya, July 2017. Data transcription, validation, and initial high-level interpretation of meaning was performed within the interview team immediately after each interview. Each interview (lasting 45–75 min) was carried out by at least two researchers, with one researcher interviewing the athlete and the other taking notes and asking follow-up questions. Data transcription, validation, and initial high-level interpretation of meaning was performed within the interview team immediately after each interview. 

### 2.3. Data Analysis

A thematic narrative analysis was used to structure the interview data as narratives [22]. The analysis was performed in a process where contextual and narrative themes were structured, analysed as stories, and finally represented as narratives [32]. The thematic component focused on analysis of the “whats” of individual stories and sought to identify common themes among these [23]. The themes identified in each illness or injury story were separated into (i) descriptive themes, i.e., an orientation (introduction of place of illness or injury, time, and persons involved), and a focus (the specific health problem), and (ii) narrative themes, i.e., an interpretation (how the health problem was understood), and a resolution of the events (how the health problem was managed) [33]. While the descriptive themes were developed using a “coding reliability” approach to thematic analysis, the narrative themes were created using reflective thematic analysis [24]. While the creation of the descriptive themes was based on the semantic meaning of the interview data, the development of the reflective themes used both the semantic and latent meaning. The individual stories were divided into these themes and the themes compared between stories. Content and structure of stories with similar meanings (evaluation elements) were then examined within the interview contexts, and the story meanings and functions were described. The evaluation elements were composed into the final narratives. These narratives made use of both quotations and personal stories. Strategies used to clarify the meaning of the stories included the use of direct quotations and variations in the tone or form of presentation. Two of the authors (T.T. and K.F.) coordinated the data analysis. One author (K.F.) developed the descriptive themes and another author (T.T.) created the narrative themes. Thereafter, the narratives were developed by the main author (T.T.) in interaction with a medical anthropologist (J.N.). All authors contributed to the final presentation. 

## 3. Results

### 3.1. Descriptive Themes

The young athletes sustained their health problems in a wide variety of settings, ranging from their homes and local sports facilities in rural areas to competition venues and boarding schools (Table 1). A broad range of health problems were recounted. Many of the athletes had felt injury pain develop gradually when training. But the health problems also included injury pain with sudden onset, e.g., from hamstring strains and ankle distortions. Regarding illnesses, the athletes mostly reported acute illnesses, e.g., influenza, chickenpox, and appendicitis. A few athletes had suffered long-term illness such as an eating disorder, malaria, asthma, and mood disorders. The medical support provided to the athletes varied. Some athletes had specialized physicians, nurses, physiotherapists, physiologists, and psychologists connected to their sports clubs, sporting schools and national teams. Most athletes had attended public hospitals and private clinics for management of their health problems.

### 3.2. Narrative Themes

Several athletes described believing that their injury was caused because they had kept on training with pain to qualify for competitions and national teams (Table 2). Some athletes also stated that their parents wanted them to continue to train while injured to avoid compromising their future athletic career. Other causes to injury incidents described by the athletes were defective or maladjusted equipment, lack of warm-up, and poor use of event-specific technique. Most athletes had some basic knowledge of how to avoid health problems. This included listening to the body, avoiding overtraining, decreasing training when experiencing pain and eating good food to help prevent injuries and illness. The athletes had less knowledge about how to treat health problems. Some athletes used social media sources, such as YouTube, to gain knowledge about training, medicine, and psychology. Most of the young athletes stated that they did not have any special sport specific health insurance. Many athletes from Africa and Asia did not have any health insurance of any kind. The European athletes often had a general health insurance and had received pre-participation cardiac and musculoskeletal examinations through their national sports federation. Some athletes also had contact with privately funded sports physiotherapists and biomechanical consultants on a regular basis, while others did not have any possibilities to pay for basic medical support at all.

#### 3.2.1. “I Got the Flu Due to Cold Weather”—The Narrative of Insufficient Health Literacy

The young athletes, regardless of background, had difficulties making sense of symptoms of ill health on their own. An African sprinter described escalating knee pain and swelling that had troubled him for about a year. He explained: “When it gets cold, it affects my knee. I need to keep the knee warm”. Due to the severity of his symptoms, the athlete was provided a clinical examination by the medical team at the world championships, to which he had qualified. The examination showed that the athlete suffered from Osgood Schlatter disease. Due to ignorance of such a condition, the sprinter had never consulted a clinical professional for this problem. The difficulty in interpreting symptoms also complicated illness. Another African sprinter tried to understand why he had caught an influenza-like illness. However, his reasoning about its etiology did not include infectious agents: “I believe that I got the flu due to cold weather”. Accordingly, the runner had not taken precautions, such as social distancing or increased hand hygiene, to protect himself after his friends and family became infected.

#### 3.2.2. “The Federation Provides No Medical Support”—The Narrative of Unequal Medical Service Access

The young athletes generally relied on the general healthcare services for treatment of illnesses they could not manage on their own. An African middle-distance runner reported that he had suffered influenza symptoms with fever and sore muscles for more than a week when the symptoms spread to the chest. The runner then worried that he had caught pneumonia. “I spoke with my coach and asked him for help to see a physician. It is free to see a doctor for me [healthcare is tax-financed]”. The coach arranged that the athlete was examined by a physician. “My worry got relieved”.

The adolescent athletes did not normally have regular access to sport medical services for diagnosis and treatment of sports injuries. Instead, they were dependent on to their personal network. The resources in these networks varied considerably, from physiotherapists and biomechanical laboratories to the advice of village neighbours and schoolteachers. For example, a European middle-distance runner suffered from gradually increasing bilateral lower leg pain about six months before the world championships. The runner told his coach and physiotherapist about the problem, but they could not explain the pain. Instead, they referred him to a biomechanics laboratory for video recordings and “analyses by a computer program”. According to the laboratory report, the runner’s upper body movement was constrained, which negatively influenced his running. Based on this information, the coach and the runner reduced the training load and adjusted his running technique. In comparison, an East African middle-distance runner who frequently suffered lower-leg pain, this time again suffered pain some six months before the championships. The African runner stated that he did not understand what caused the pain. He also said that he had no one to turn to when he experienced injury symptoms. The runner explained that he always “feels bad when injured”, because he then perceives himself as helpless. “The federation provides no medical support”, he said frankly. He never had a physiological test or medical screening. The runner affirmed that the unresolved pain problem led him to “not perform well at the world championships”.

#### 3.2.3. “We Cut through the Skin and Applied Herbs”—The Narrative of Inept Sports Injury Management

When having been provided access to a healthcare facility, the illness treatments received were in general described to have been effective. An Asian race walker described an episode when he fell ill with knee and elbow pain. The illness developed as a sequel to an infectious disease, well known in his community, distributed by mosquitos. He went with his coach to the local private hospital, where analyses of blood samples were made, and medications given. His family paid out-of-pocket for these services. The race walker explained that there was no public healthcare available in his community, and that he did not receive any support from his sports club or the national athletics federation. Nevertheless, the treatment at the hospital was successful and he was able to return to training. 

Regarding sports injuries, there seemed to exist major variations in the quality of the treatments provided to the young athletes, ranging from traditional remedies and rest compared with biomechanical interventions. For instance, an East African middle-distance runner living in a rural village suffered continued pain problems following repeated ankle sprains. Due to that no facilities for treatment of sports injuries were available in his community or nearby, his only option was to turn to traditional healers nearby, who provided him herbal remedies (Table 3). However, upon closer examination, none of the young athletes in this study reported that they, or their parents or coaches, had asked for, or been shown, evidence on the effectiveness of the sports injury treatments they had received. A typical example was a European runner who had suffered pain inside his lower legs during training (Table 4). Although he was provided several treatments, his symptoms and the rationale for his treatments were never explained to him. The remedy for this runner’s injury was eventually reduction of training load.

#### 3.2.4. “The Little Engine That Could”—The Narrative of Self-Sufficiency and Personal Determination 

When asked to summarize their experiences of medical support, adolescent athletes often repeated that they had to manage their health issues on their own. As children, the athletes had received support from parents, other relatives, and community members close to them. However, when they reached adolescence and increased their training, their needs for medical services extended beyond what parents, schoolteachers, and others nearby could provide. For example, an African middle-distance runner grew up with his mother in a remote rural village. His only coaches were local schoolteachers without any specific background in sports, and he self-managed his health problems. The runner qualified for the youth world championships with an outstanding personal best in his event, despite having suffered repeated injuries (Table 5). However, an undisclosed and untreated overuse injury sustained at pre-event camp forced him to break off championships participation although he reached the finals. A similar story was told by a European endurance athlete who reported that she had fallen ill with stomach pain while staying at a sports boarding school. The lack of attention paid to her symptoms there led to a delayed diagnosis of appendicitis, followed by a complicated clinical intervention and rehabilitation process (Table 6). Although the parents had been concerned, they had not had the means to provide the athlete the support she needed. Meanwhile, the coaches and managers seemed to have been uninterested in her health and well-being off the track. Such “the little engine that could” narratives that stress self-sufficiency and individual determination as key to success were common among the adolescent athletes. The issues that appeared to force the young athletes into a premature adulthood could extend beyond illness and injury management per se, as displayed in the case of an African runner (Table 7). Despite his young age, he stayed alone in a rural village, where he had a coach who he had to pay on his own. The athlete said that he often was ill due to poor food in his village and that no medical services were available. Despite their own poverty, his village neighbours tried to support him with groceries and other supplies. In return, the young runner stated that the possibility to look after his home village in the future was a primary career motivation for him. 

## 4. Discussion

This qualitative study aimed to investigate experiences from medical service provision among high-level adolescent athletes from Africa, Europe, and Asia. The rationale was to contribute to the development of a medical service stewardship that ensures emerging professional sportspersons efficient and equal medical support worldwide. The results exposed noteworthy insufficiencies and disparities concerning service access (geographic distance, costs, availability of insurance) and quality (trained clinicians, use of efficient therapies). The athletes’ accounts thus uncovered a disquieting dissonance between present practices and ethical standards in competitive youth sport pertaining to health safeguarding and equality [34,35]. The findings are above all striking from a health policy perspective because considerations of equity and justice should be key concerns when assessing medical services across regions and nations [36]. It can be noted that only within-nation, not international, socioeconomic disparities were addressed in a recent review of knowledge on ‘the development of the world’s best sporting talent’ [37]. Such bracketing of inequity at the global level may render populations of already disadvantaged adolescent athletes not only less competitive but further at risk.

### 4.1. Moving Beyond Self-Sufficiency and Individual Determination

About one third of adolescent competitive athletics athletes suffer at least one significant injury in one year [18]. Although competing at the highest level, the adolescent athletes in this study generally lacked the knowledge to independently interpret injury and illness symptoms and they were seldom guided to a diagnosis and evidence-based treatment by the adults close to them. It can be argued that these “invisible” shortcomings were a hazard to the adolescent athletes’ long-term health and lessened their opportunity to be competitive in their sport as adults. This sets the stage for athletes who come from places with more medical resources “to do better”, theoretically at least, in competition than the rest of their peers. Further, the acceptance of the idea that to endure hardships is necessary for succeeding in sports can be strongly disadvantageous among young top-level athletes, particularly for those who have less opportunities to access high-quality medical services. By endorsement of athlete development processes that emphasise individualism and personal toughness, Ref. [38] international sports bodies thus can evade the responsibility for working towards equity in health safeguarding among adolescents highly talented in sports. The endorsement of “the little engine that could” narratives among young athletes is furthermore problematic because also outside sports, adolescents in many regions of the world often lack reliable adults for consultations about their health, [39] and they often face larger barriers than children and young adults in accessing clinical services [1]. For instance, access to financial support, ineligibility for tax- or insurance-based funding schemes, legal requirements for parental consent, and societal norms may cause adolescents to delay or avoid seeking medical services [40]. Current international health policies emphasize that all adolescents should have an equal opportunity to attain their full health potential and that none should be disadvantaged [1,4]. These findings imply that public health agencies should increase their involvement with adolescents participating in top-level sports.

### 4.2. Public Health Involvement with Adolescent Top-Level Athletes

This study revealed that young talented athletes often have to put up with critical shortcomings in the handover of responsibility for medical support from parents and local communities to sports organisations. Because athletics is a global sport, remedial actions addressing top-level athletes must include equity and sustainability and be administrated through transnational governance [41]. Transnational sport governance has been defined as the ‘global regulation of sport and sport competitions, especially those involving nationally-representative athletes and teams’ [42]. The global population of adolescents aspiring to qualify for the World Under-20 Championships in athletics can be estimated to about 3000–4000 athletes per birth-year cohort distributed to more than 200 nations. These results suggest that a suitable basis on which to configure governance of medical service provision to these athletes is for public health agencies and sports bodies to embrace a common stewardship, i.e., that they share the responsibility for financially efficient medical service outcomes and an ethical use of available resources [43]. A partnership also with a global health agency would here assure the competence and authority to implement an accessible program with credibility also outside sports. For financing, the player drafting system in North American team sports can be used as a model. The drafting system has parallels with other equally attractive economic schemes that combine financial considerations with ethics and redistributive social protection mechanisms [44]. As a starting point for drawing up goals and strategies for the service, the Adolescent Health Services Barriers Assessment (AHSBA) tool [45] can be used to identify the categories of young athletes being left behind. The use of AHSBA tool could also foster co-operation between sports bodies and healthcare providers at all levels (from micro to macro). For the service introduction in practice, distributed service models adapted for adolescents that take advantage of modern information technology can be employed [46,47]. These models, which can vary nation to nation, region to region, should all stress that enhanced access and adolescent participation in the service design are important features to be implemented. 

### 4.3. Study Strengths and Limitations

The study has strengths and weaknesses that need to be considered when interpreting the results. The narrative thematic method is well established for the unfolding of an individual athlete’s experiences over time and to situate personal stories in relation to broader social and environmental contexts [48]. In accordance with the research task at hand, two approaches to thematic analysis were applied. The coding reliability approach was used for creation of themes describing the context of the health problems, while the central analysis, the development of the narrative themes, was based on the reflective approach [24,49]. Coding of the latter themes was open and organic, with iterative theme development and sparse use of a coding framework. Regarding potential weaknesses, it must be acknowledged that the study sample was limited in size. However, because the study was performed using qualitative methods, it is not meaningful to quantify the results. For instance, for example the differences within and between global regions in the proportions of athletes having access to certain health resources. Instead, the study can be positioned within the framework of intersectional generalisability, which combines “theoretical” and “provocative” generalisability [50]. This means that lessons can be learnt about disadvantages and neglect moving from one context to another and researchers can be used to rethink “the possible” and begin to investigate arenas not yet canvassed or imagined. Moreover, although the study included participants from countries at both ends of the Human Development Index, it was restricted to only English-speaking athletes. There may be further cultural and other differences in how injury and illness are dealt with among young athletes both within the countries and continents covered by the study, and in other world regions. It is also likely that the English-speaking athletes participating in the study had received more formal education and were better off than their non-English-speaking peers in their respective home setting. When planning the study, we recognised that data saturation was not the ideal term to use in association with reflexive thematic analysis [51]. We had interviewed 14 athletes when an in-situ decision was made that the collected data were adequate in terms of richness and complexity for addressing the research question. The data saturation concept and sample size estimations in connection with application of different versions of thematic analysis warrant further research.

## 5. Conclusions

Most previous studies of medical service provision to adolescent high-level athletes originate from the U.S. The experiences of adolescent top-level athletics athletes from Africa, Europe, and Asia reported in this study disclose serious insufficiencies in the availability of medical services at critical periods in their careers. The results support the development of a global stewardship model for medical service delivery to adolescent top-level athletes based on partnerships between public health agencies and sports bodies. This would provide the social and fiscal capital necessary for upholding the ethical standards for top-level youth sports as well as for the equal provision of medical services to adolescents across the world. Further inter-disciplinary research on the topic is warranted.

## Figures and Tables

**Table 1 ijerph-18-07278-t001:** Descriptive themes.

Descriptive Themes	Subthemes
Settings	Home or local sports facility Competition venue Boarding school
Health issue (lay diagnosis)	Traumatic injury Overuse injury Acute physical illness Long-term physical illness Mental illness
Health service provided	Sports physiotherapist Sports physician (nurse) General healthcare Traditional medicine

**Table 2 ijerph-18-07278-t002:** Narrative themes.

Narrative Themes	Subthemes
Hazardous sports practice	Training when unwell (weight loss, tiredness) Training with pain Training forced by coach or family
Actionable prevention knowledge	Avoid overtraining (listen to the body) Appropriate nutrition
Dissatisfied health service need	No clinician available Did not dare to talk to the coach Clinician did not care Treatment did not work
Socioeconomic inequity	Personal support resources Sports medical insurance Equipment supply Nutrition
Adult leadership failure	Poor career guidance (dual career prospect) Lack of health counselling

**Table 3 ijerph-18-07278-t003:** Story of East African middle-distance runner. Male, age 17 years.

The runner suffered gradually increasing pain from his right ankle after having rolled that ankle several times. His athletic talent had been recognized in school competitions, and he had qualified for admission to a well-known sports college in the capital city. However, the pain increased suddenly when the athlete rolled the ankle again at an inter-school championship, which forced him to disrupt training and competition. The athlete describes the events that then followed: *“My coach told me to just rest, but I have no trust in him. There are no medical services available in my village. My father got very concerned that the injury would inflict my possibilities to study at the sports college. He therefore helped me to use herbal remedies. This is traditional medical knowledge passed down through generations in the village. We cut through the skin and applied the herbs”.* The athlete found the rest and the herbs were efficient, but it took four months before the pain subsided.

**Table 4 ijerph-18-07278-t004:** Story of European middle-distance runner. Male, age 17 years.

The runner suffered gradually increasing pain from the inside of his lower legs. He had increased his training load from 10 to 14 h a week after having been admitted to a sports school. His coach recommended him to see a physician, who diagnosed the problems as shin splints. The runner was referred to a physiotherapist for treatment. The physiotherapist told him that his hip muscles were too weak and prescribed a special strength training program. This program did not help him. After 2 months, the runner was given another exercise program from a teammate who had had similar problems. The program consisted of rubber band exercises focused on strengthening the muscles in the lower leg (“*around the shins*”). He also met another physiotherapist living close to his home, who provided laser and acupuncture treatments. These treatments did not help either. The runner finally decreased his training load after discussion with the coach. There were no competitions in the winter when he got his problems, and the runner could again slowly increase his training load. The runner established that “*my body was not used to this extra amount of training*”. After four month he was back in full training.

**Table 5 ijerph-18-07278-t005:** Story of East African middle-distance runner. Male, age 17 years.

The runner experienced pain and swelling of his right lower leg about eight months before the world championships. He trained alone in his home village, with no coach or medical support available. The athlete tried self-treatment of the pain by immersing the leg in hot water. He managed to win the regional trials for the national cross-country championships, but could not compete in the finals due to pain. The runner then decided to try shorter middle-distance track events, even though he never had run in track shoes or competed on track before. Despite that the painful swelling of his leg remained, he won a national middle-distance running trial by a wide margin and was selected for the world championships team. At the pre-championships camp, the runner initially blamed symptom continuance on his new track shoes and tried to reduce training on track. When he eventually turned to a team coach, the coach “*did not listen*” to him. The runner then described his problems for the team physiotherapist, who was unable to provide a diagnosis. The runner and physiotherapist tried massage and pain medication. This treatment was initially successful. At the world championships, the runner managed to advance through the semi-finals. However, the pain increased after the race, and he was unable to compete in the finals.

**Table 6 ijerph-18-07278-t006:** Story of European endurance athlete. Female, age 17 years.

The athlete stayed at a boarding house in a sports school. She fell ill with stomach pain, but her teachers and coaches did not pay attention to her complaints. When her agony became almost unbearable, she called her mother who helped her to access a private hospital in her hometown. The athlete was diagnosed with appendicitis and went through surgery. After four weeks, she was back in training. However, the pain returned. It was found that the infection had spread to several organs. The athlete underwent surgery a second time and had to stay 3 weeks in hospital. At this time, she was worried that the illness had ended her career. She had lost 10 kg in weight, and could only start easy training. After the illness episodes, the athlete moved to a new boarding house and sports school, where the coaching and medical facilities were “much better, with several specialized doctors and physios”. The athlete trained hard after the second surgery, and she managed to qualify for the world championships. *”I fought to be here, I dreamt to start here”, as she put it*.

**Table 7 ijerph-18-07278-t007:** Story of East African middle-distance runner. Male, age 17 years.

The runner lived alone in a village. His father was deceased and his mother lived in another region of the country. He met a coach twice a week, but mostly he trained by himself. When he did well in competitions and won some prize money, the coach took most of it. The village neighbours tried to support the runner with food and material resources as best as they could. The athlete told that he always was worried about suffering an injury or illness, due to that he had ”*no good food available*” and did not eat well enough to withstand a heavy training load. He also owned only one pair of worn-out running shoes and one pair of track shoes for competition and could not afford to buy new pairs. The athlete’s recognition of the association between his personal health, ambition, and sports performance, and the resources available in the local community was displayed when he stated that he wanted to succeed as a runner in order to help ”*support the poor*” in his home village.

## Data Availability

No additional data are available.

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
