# Peer review of "‘The Little Engine That Could’: A Qualitative Study of Medical Service Access and Effectiveness among Adolescent Athletics Athletes Competing at the Highest International Level"

_ijerph, 2021, doi:10.3390/ijerph18147278_

Round 1

Reviewer 1 Report

Thank you for the revised version of the manuscript “A little engine who could” regarding medical services to adolescent athletes. It has been improved; however, some comments are following below.

The interviews were conducted 2017, are the answers and the results still relevant and valid?

The manuscript is not discussing important parts of a qualitative study such as trustworthiness, credibility, transferability, dependability and conformability.

Your aim was to investigate experiences of medical service among high-level adolescent athletics (track and field) athletes from three continents. Your results showed differences in costs and access to medical service. It is obvious that the variations can be experienced as unfair. But it is still unclear how this study contributes to a change and how your conclusion and suggestion can be realistic. Telling the athletes’ stories about injuries and pain are interesting but it is hard to understand that medical service worldwide should be considered based on partnerships between public health agencies and sports bodies. It is hard to understand how this study design justify your conclusions. Most people involved in elite sport know these differences, so what is the knowledge gap and the contribution of this this study?

Please highlight the news you provide.

Author Response

Thank you for the revised version of the manuscript “A little engine who could” regarding medical services to adolescent athletes. It has been improved; however, some comments are following below.

Authors’ response: We thank the reviewer for the comments that indeed have helped us to improve the paper.

The interviews were conducted 2017, are the answers and the results still relevant and valid?

Authors’ response: Thank you. Not the least due to the ‘standstill’ during the Covid-19 pandemic, we find the results to be up-to-date and highly relevant. The submission of the paper was delayed, first, by unexpected circumstances among the authors and then by the outbreak of the pandemic.

The manuscript is not discussing important parts of a qualitative study such as trustworthiness, credibility, transferability, dependability and conformability.

Authors’ response: Thank you. Trustworthiness, credibility, transferability, dependability and conformability are terms applicable to the coding reliability version of thematic analysis. In this study, a reflective thematic analysis was applied. The results are presented and discussed accordingly (for more background, see references [24,49]).

Your aim was to investigate experiences of medical service among high-level adolescent athletics (track and field) athletes from three continents. Your results showed differences in costs and access to medical service. It is obvious that the variations can be experienced as unfair. But it is still unclear how this study contributes to a change and how your conclusion and suggestion can be realistic. Telling the athletes’ stories about injuries and pain are interesting but it is hard to understand that medical service worldwide should be considered based on partnerships between public health agencies and sports bodies. It is hard to understand how this study design justify your conclusions. Most people involved in elite sport know these differences, so what is the knowledge gap and the contribution of this this study?

Authors’ response: Again, thank you. Although neglect of health service provision to adolescents has been highlighted in the general literature on health sciences research and public health, we find that the topic has not been sufficiently highlighted at the international level in sports medicine (Timpka 2008). Although the project we suggest is challenging, health service development for adolescents has been shown to be successful, as displayed, for instance, in the Australian Headspace program (Rickwood 2019).

Timpka T, Finch CF, Goulet C, Noakes T, Yammine K; Safe Sports International Board. Meeting the global demand of sports safety: the intersection of science and policy in sports safety. Sports Med. 2008;38(10):795-805.

Rickwood D, Paraskakis M, Quin D, Hobbs N, Ryall V, Trethowan J, McGorry P. Australia's innovation in youth mental health care: The headspace centre model. Early Interv Psychiatry. 2019 Feb;13(1):159-166. 

Please highlight the news you provide.

Authors’ response: The Conclusions have been thoroughly revised to highlight the take-away messages from the study.

Reviewer 2 Report

This is an overall well written and as far as I can see well performed qualitative study of medical service access and effectiveness among adolescent athletics competing at the highest international level. The topic is important as it has not been studied in deep earlier and it contains a lot of interesting facts.

The paper is unfortunately somewhat difficult to read for three reasons. The main reason is the disposition of the result section. The s.c. tables are not positioned closed to the matching text. As an example one has to read another 3 pages until you find the reference to table 3 on page 5. Secondly it is odd to name the stories as tables. Why not just call them Story 1, 2, 3 a.s.o. Thirdly matching texts are often a mixture of direct results and examples that just as well could have been presented as short stories.

Please describe better how the decision of the saturation of the sample size was made.

As far as I can see there is no research ethical consent and no ethical discussion!

minor comments:

Table 1 should be placed in the result section.

page 2 line 63: What about middle income countries?

page 3 line 121-124 are repeating the text just above.

page 3 line 132:  there is an i). Should there also be ii) iii) a.s.o.?

page 3 line 136: "coding reliability". Reliability is a term for quantitative research. In qualitative research you normally talk about "dependability". Please check this also in the discussion on page 9 line 368.

page 7 line 262: where does the concept "a little engine that could" come from? Is it from a story or is it a concept created by the authors? When it comes back in discussions on pages 8 line 281 and 322 it is not clear what it means.

page 8 line 291: I suppose that the discussion section starts from here?  Needs a heading.

Author Response

This is an overall well written and as far as I can see well performed qualitative study of medical service access and effectiveness among adolescent athletics competing at the highest international level. The topic is important as it has not been studied in deep earlier and it contains a lot of interesting facts.

Authors’ response: Thank you for the comments.

The paper is unfortunately somewhat difficult to read for three reasons. The main reason is the disposition of the result section. The s.c. tables are not positioned closed to the matching text. As an example one has to read another 3 pages until you find the reference to table 3 on page 5. Secondly it is odd to name the stories as tables. Why not just call them Story 1, 2, 3 a.s.o. Thirdly matching texts are often a mixture of direct results and examples that just as well could have been presented as short stories.

Authors’ response: Thank you for the comment. We are willing to make all the changes suggested, but have not implemented them yet. We agree with the reviewer, but also find that the stories may be easier to read as tables. We kindly leave to the copy editors to decide whether or not to convert the tables to “story” sections in the running text.

Please describe better how the decision of the saturation of the sample size was made.

Authors’ response: In the revised manuscript, we describe the approach to saturation, as it is used in reflective thematic analysis, in the methods section and provide a comment and a reference ([51]) to an article where the decision is discussed at length in the limitations section.

As far as I can see there is no research ethical consent and no ethical discussion!

Authors’ response: The ethical considerations are reported at the end of the paper (after the main text).

minor comments:

Table 1 should be placed in the result section.

Authors’ response: Thank you. We fully agree. Please see comment on copy-editing and layout above.

page 2 line 63: What about middle income countries?

Authors’ response: Thank you. Middle-income countries were of course also included. A revision has been made.

page 3 line 121-124 are repeating the text just above.

Authors’ response: Thank you. A revision has been made.

page 3 line 132:  there is an i). Should there also be ii) iii) a.s.o.?

Authors’ response: Thank you. Yes, ii) follows two rows below i).

page 3 line 136: "coding reliability". Reliability is a term for quantitative research. In qualitative research you normally talk about "dependability". Please check this also in the discussion on page 9 line 368.

Authors’ response: Thank you for this pertinent remark. Yes, we agree. We used the "coding reliability" approach (see [24]) to thematic analysis only for the initial structuring of the data on the context of the health service need. Our ambition has been to follow the notions used in [24] throughout the paper.

page 7 line 262: where does the concept "a little engine that could" come from? Is it from a story or is it a concept created by the authors? When it comes back in discussions on pages 8 line 281 and 322 it is not clear what it means.

Authors’ response: Thank you for the comment. Yes, the expression “the little engine that could” comes from a famous story for children by Watty Piper. This tale tells the story of a small engine that succeeds in pulling a train over a mountain while repeating the motto: "I-think-I-can". The underlying theme is notably similar to that found among the adolescent athletes in this study. We find that the tale, as a universal metaphor about issues that a young individual may confront, highlights a critical ethical question in the athlete development process, i.e,. what level of responsibility that is fair to allocate to the young sportspersons themselves.

page 8 line 291: I suppose that the discussion section starts from here?  Needs a heading.

Authors’ response: Thank you. This is correct. A revision has been made.

Reviewer 3 Report

Rationale of the study should be more clear and stronger. It'll help to better develop the objective.

Methods: how about language of data collection. As there are responses from different countries/continents, how the authors overcame or tackled the language barriers? As  they had selected only English speaking athletes,  it'd be better to bring the language issues under method discussion and mention the probable biases in the study results.  Even the authors selected participants from both end of HDI index, but intra country variation for language is critical issue. 

Author Response

Rationale of the study should be more clear and stronger. It'll help to better develop the objective.

Authors’ response: We do agree. A thorough revision has been made of the Introduction in order to make the rationale more clear and stronger.

Methods: how about language of data collection. As there are responses from different countries/continents, how the authors overcame or tackled the language barriers? As  they had selected only English speaking athletes,  it'd be better to bring the language issues under method discussion and mention the probable biases in the study results.  Even the authors selected participants from both end of HDI index, but intra country variation for language is critical issue. 

Authors’ response: We again agree. Text has been added to the limitations section pointing out the English-speaking athletes may not be entirely representative even for the population of athletes from the own country.

Round 2

Reviewer 1 Report

Thank you for a well written paper! As I can see, relevant changes have been made in this version.

This manuscript is a resubmission of an earlier submission. The following is a list of the peer review reports and author responses from that submission.

Round 1

Reviewer 1 Report

Thank you very much for offering the possibility to review the paper ijerph-1173474 titled “‘A little engine who could’: an international perspective on medical service access and quality among adolescent top-level athletes in athletics (track and field)”

Just reading the title, the paper is interesting and seems to be an interesting topic. However, when we delve deeper into it, we discover certain shortcomings for a journal of the quality and impact of the IJERPH.

A number of recommendations and the justification for this statement are given.

The title is somewhat vague and can be confusing. Among other things, it would be interesting to include that it is a qualitative study.

The abstract is correct, although information on the study participants and a summary of the conclusions are missing. Include in the keywords: "qualitative study".

The introduction is seriously lacking. Firstly, it is too short. The authors should make it longer, as the topic of study proposed by the authors allows them to do so.

As such, it is somewhat confusing. I recommend the authors to restructure it and to give a global view and emphasise the problematic of the study.

Furthermore, the authors cite studies that are not up to date and some of them are of low scientific quality. The authors should do a literature review in different databases in the last 5 years (2017-2021).

The aim should be the last thing to appear. Before this point, the problem and the "why" of the study should be justified (the authors do this but it is not well structured and can be confusing). However, the research questions formulated by the authors, which precede the objectives, still need to be included.

Another very important aspect, but at the same time with serious limitations, is the materials and method.

Firstly, I recommend authors to establish sub-sections for the readers' understanding. For example: "study design", "participants", "procedure", etc.

In this sense, the information that the authors have included in the supplement should be synthesised and linked in this section, as it is important that they appear in this section.

Socio-demographic information on the participants is missing. Distribution by disciplines, gender, continents, countries, average age, etc.

As I indicated earlier, the information in the supplements should be included and structured. It is interesting that codes, categorisations, processes, etc. appear. These aspects need to be defined.

Have the authors carried out the data analysis with any statistical software? ATLAS, AQUAD, etc.

The section on the results, according to the authors' approach, is correct. However, it also has limitations. The results and statements made by the authors should be reflected in the percentages of coincidence, thus facilitating the readers' understanding.

It is recommended that for each of the results (subsections), the authors use a statistical programme and include the graphs they provide to establish an analytical basis for consistency.

Were these results extracted by one author or by all? What is the concordance percentage? This aspect should be referenced in the methodology when describing the coding.

If instead of using tables, the authors were to make graphs with all participants, it would facilitate the understanding of the study. It would also provide greater scientific rigour.

In the discussion section, you should start with a paragraph stating the objective of the study and compare it with other research of similar nature or characteristics. This should be followed by a paragraph stating what is new to the scientific community that has not already been done.

Similarly, a literature review of the last 5 years is missing. These should be reviewed throughout the document. For the quality of the journal, it would be fine to include information on 10-12 papers from the last few years.

For the conclusions, could practical applications of the study be established?

The references do not conform to the journal's standards and should include all those that are updated.

Overall, the study idea is interesting but does not seem to me to be novel or well-founded. As the authors say, the sample is too small and lacks a consistent scientific method.

Reviewer 2 Report

Thank you for conducting the study and writing the manuscript “A little engine who could” regarding medical services to adolescents athletes. It is well written, however, some comments are following below.

The section on page 2, lines 48-57, discuss health in general, and do these suggestions and goals relate to the need for sport medicine support? In that case sports medicine might consider as luxury and general health should be prioritized.  Consider to condense this section.

Page 2, lines 63-65, you have both an aim and a purpose, Please consider to collapse them.

The method part is lacking sound inclusion and exclusion criterion.  

Page 2, line 84, (and other places) you wright that you reached saturation. Is that need in this type of analysis and method? See Virginia Braun & Victoria Clarke (2019): To saturate or not to saturate?

Questioning data saturation as a useful concept for thematic analysis and sample-size rationales,

Qualitative Research in Sport, Exercise and Health, DOI: 10.1080/2159676X.2019.1704846.

This manuscript is lacking some of the important parts of a qualitative study such as trustworthiness, credibility, transferability, dependability and conformability.

Page 3: results: Please explain the themes that emerged from your analysis.

Page 3, line 108: please check the sentence starting the with The examination…

The tables, especially table 1, is very much the same as the text. Do they really contribute to the result?

Page 5, line 198: you refer to the case of an African runner. It is hard to follow who you mean. Please refer to a table or explain more.